# Immunomodulatory Effect of *Lactobacillus reuteri* (*Limosilactobacillus reuteri*) and Its Exopolysaccharides Investigated on Epithelial Cell Line IPEC-J2 Challenged with *Salmonella* Typhimurium

**DOI:** 10.3390/life12121955

**Published:** 2022-11-22

**Authors:** Zuzana Kiššová, Ľudmila Tkáčiková, Dagmar Mudroňová, Mangesh R. Bhide

**Affiliations:** 1Institute of Immunology, University of Veterinary Medicine and Pharmacy in Kosice, 041 81 Kosice, Slovakia; 2Laboratory of Biomedical Microbiology and Immunology, University of Veterinary Medicine and Pharmacy in Kosice, 041 81 Kosice, Slovakia

**Keywords:** IPEC-J2, *Lactobacillus reuteri*, *Limosilactobacillus reuteri*, exopolysaccharides, probiotics, *Salmonella* Typhimurium, cytokines, TLR

## Abstract

The gastrointestinal tract is the largest and most complex component of the immune system. Each component influences the production and regulation of cytokines secreted by intestinal epithelial cells. The aim of this study was to see how the probiotic strain *Limosilactobacillus reuteri* L26 and its exopolysaccharide (EPS) affect porcine intestinal-epithelial cells IPEC-J2 infected with *Salmonella* Typhimurium. The results revealed that *Salmonella* infection up-regulated all studied pro-inflammatory cytokines such as TNF-α, IL-8, IL-6 and TLR4, TLR5 signaling pathways, while decreasing the expression of TGF-β. An immunosuppressive activity was found in EPS-treated wells, since the transcriptional levels of the studied pro-inflammatory cytokines were not increased, and the pretreatment with EPS was even able to attenuate up-regulated pro-inflammatory genes induced by *Salmonella* infection. However, there was a significant increase in the expression of mRNA levels of IL-8 and TNF-α in L26-treated cells, although this up-regulation was suppressed in the case of pretreatment. The immunoregulatory function of *L. reuteri* was also confirmed by the increased level of mRNA expression for TGF-β, a known immunosuppressive mediator. The most relevant finding of this ex vivo study was a case of immunity modulation, where the probiotic strain *L. reuteri* stimulated the innate immune-cell response which displayed both anti- and pro-inflammatory activities, and modulated the expression of TLRs in the IPEC-J2 cell line. Our findings also revealed that the pretreatment of cells with either EPS or live lactobacilli prior to infection has a suppressive effect on the inflammatory response induced by *Salmonella* Typhimurium.

## 1. Introduction

Probiotic bacteria are well known for their ability to modulate the mucosal immune system, which benefits the host organism. They have a significant impact on homeostasis, inflammatory response, and immunopathology suppression; however, these activities vary from strain to strain [1]. Lactobacilli can produce a variety of extracellular polymers known as exopolysaccharides (EPSs), which have a wide range of structures, composition, and biological function [2]. EPS are either attached to probiotic bacteria cell walls or secreted into the environment [3]. These biopolymers play pivotal roles in the protection of bacteria and adhesion to host cells; however, they have not only positive effects on their producers [2], but also immunomodulatory effects on the gut mucosal immune system [4]. Through interactions with immune cells, such as macrophages and dendritic cells, probiotic EPS are involved in innate-immune-modulation processes, as well as in adaptive immune responses which increase T- and B-cell proliferation [5]. It has been even proposed that the health benefits of probiotic bacteria for the host can be attributed to the production of EPS [6].

Intestinal epithelial cells (IECs), the first line of defense, prevent various pathogens and their toxins from entering the systemic circulation [7]. Different membrane pattern-recognition-receptors (PRRs) are found in epithelial cells, for example Toll-like receptors (TLRs), which serve to recognize invariable highly conserved molecular-structures of microorganisms [8]. Since inflammation is known as a crucial part of *Salmonella enterica* infection, effector molecules secreted by *S. enterica* and the detection of MAMPs (Microbe-associated molecular patterns) initiate the host inflammatory-responses via TLR, followed by the production of pro-inflammatory cytokines such as IL-8, TNF-α, IL-1β and IL-18. The change in the intestinal microflora which occurs after the *S. enterica* infection in the host, create favorable conditions for the growth of *Salmonella* itself [9,10].

In a previous study, we found that EPS isolated from the probiotic strain *L. reuteri* L26 was able to attenuate the inflammation evoked by the *E. coli* infection in the IPEC-1 cell line. We also found that EPS suppressed the expression of genes activated by the heat-labile and heat-stable toxins of ETEC, which can lead to electrolyte loss in the gut [11]. The aim of the present study was to characterize the response of the porcine jejunal-cell-line (IPEC-J2) to a *Salmonella* Typhimurium infection, by detecting the mRNA expression level of pro-inflammatory cytokines such as TNF-α, IL-6, and chemokines such as IL-8. We further aimed to investigate the effect of a live probiotic strain and, separately, its EPS at the cellular level.

## 2. Materials and Methods

### 2.1. Bacterial Strains

*Limosilactobacillus reuteri* L26 Biocenol^TM^ (CCM 8616) was a kind gift from Assoc. Prof. Radomíra Nemcová, Institute of Microbiology and Immunology, University of Veterinary Medicine and Pharmacy in Košice, Slovakia. *L. reuteri* L26 (L26) was cultured in modified de Man–Rogosa–Sharpe medium (MRS; HiMedia, Thane, India) containing 10% sucrose (Mikrochem, Pezinok, Slovakia). Overnight, cultures of lactobacilli were inoculated into fresh modified MRS medium and incubated for the next 24 h at 37 °C, without shaking, prior to the experiment. The EPS extraction was carried out from the overnight cultures of L26, incubated for 48 h at 37 °C, without shaking, in modified MRS medium.

*Salmonella* Typhimurium CCM 7205 (ST) was obtained from the Czech collection of microorganisms. *Salmonella* were cultivated in Luria–Bertani broth (LB; Sigma-Aldrich, St. Louis, MO, USA) overnight at 37 °C, with constant shaking (160 rpm). The overnight cultures were inoculated into fresh LB medium and incubated for 4 h at 37 °C, with constant shaking.

In order to quantify each bacterial concentration, the optical density (OD) was measured at a wavelength of 600 nm in a Synergy HTX Multi-Mode Reader spectrophotometer (Agilent, Santa Clara, CA, USA) and confirmed by serial dilution with a determining CFU count on Mueller–Hinton agar plates. Each bacterial strain was cultivated at 37 °C, until the mid-log phase was reached, then centrifuged and washed twice with phosphate-buffered solution (PBS). Prior to addition to the IPEC-J2 cells, the bacteria were diluted in a serum- and antibiotic-free IPEC-J2 cell culture medium at a concentration corresponding to a multiplicity of infection (MOI) of 50 bacteria to a cell. 

### 2.2. Extraction and Purification of EPS

Isolation and purification of EPS from *L. reuteri* L26 for the purpose of the current experiment was performed exactly as described in our previous work [12].

### 2.3. IPEC-J2 Cell Culture

The IPEC-J2 cell line was provided by Juan José Garrido from the Department of Genetics and Animal Breeding, University of Cordoba, Spain. IPEC-J2 cells were grown and maintained in DMEM/F-12 (Sigma-Aldrich, St. Louis, MO, USA), supplemented with fetal bovine serum (5% FBS; Lonza, Switzerland), glutamine (2 mM; Biosera), epidermal growth factor (5 ng/mL; BD Biosciences, San José, CA, USA), transferrin (10 μg/mL), insulin (10 μg/mL), selenium (10 ng/mL) (Lonza), and gentamicin (50 μg/mL; Sigma-Aldrich). Cells were incubated at 37 °C in a fully humidified atmosphere with 5% CO_2,_ until confluence. Cells were seeded in 6-well culture plates (TPP, Switzerland) (1.5 × 10^5^ cells per well) 72 h before the experiment, and cultured in DMEM/F-12 medium as mentioned above, but supplemented with hydrocortisone (0.28 μM; Sigma-Aldrich) and ascorbic acid (5 μg/mL; Sigma-Aldrich) to avoid preliminary cell-activation. At 24 h prior to the experiment, the cell medium was changed to DMEM/F-12 without supplementation (without FBS and gentamicin). The cultures were regularly controlled for the absence of mycoplasma contamination [13].

### 2.4. Experimental Design—Exposure of IPEC-J2 Cells to the Different Treatments

#### 2.4.1. Exposure of IPEC-J2 Cells to *S.* Typhimurium (ST), *L. reuteri* L26 (L26), or EPS

Treatments included control (uninfected cells) and cell induction by ST, L26, or EPS (Table 1). After a three-day cultivation (37 °C, 5% CO_2_) in 6 well plates, the cells became confluent and were washed with sterile PBS. Subsequently, EPS was added to the cells in the IPEC-J2 medium without FBS in the final concentration 0.1 mg/mL, and the plates were further incubated at 37 °C, 5% CO_2,_ for 4 h. For the L26 or ST treatment, 4 h cultures (ST) or overnight cultures (L26) of bacteria were diluted to obtain a multiplicity of infection (MOI) of 50:1 (bacteria: epithelial cell), then pelleted and resuspended in IPEC-J2 medium, and plates were further incubated at 37 °C, 5% CO_2_ for 5 h (L26) or 1 h (ST). After removing the supernatant, *Salmonella*-treated cells were incubated for the next 3 h in the presence of 100 ug/mL of gentamicin (Sigma-Aldrich) to kill the remaining extracellular bacteria. At indicated times, media were removed for the later detection of IL-8 secretion, and the cells were subjected to RNA extraction for the later evaluation of chemokine or cytokine gene-expression.

#### 2.4.2. Exposure of IPEC-J2 Cells to *L. reuteri* L26 (L26) or EPS isolated from *L. reuteri* L26 (EPS) Followed by Challenge with *S.* Typhimurium

Cells were handled as indicated above on 6-well culture plates. The EPS pre-exposure experimental model included control (uninfected cells) and EPS-pretreated cells (100 ug/mL of EPS) (Table 1). The cells were incubated at 37 °C, 5% CO_2_ for 4 h, followed by the ST (MOI 50:1) challenge. The L26 pre-exposure experimental model was technically quite similar, including control (uninfected cells) and L26-pretreated cells (MOI 50:1). The cells were incubated at 37 °C, 5% CO_2_ for 5 h, followed by the ST (MOI 50:1) challenge. Cells from all treatments were then washed, replaced with fresh media containing 100 ug/mL gentamicin and incubated at 37 °C, 5% CO_2_ for 3 h. At indicated times, media were removed for the later detection of IL-8 secretion, and the cells were used for RNA extraction for later evaluation of chemokine or cytokine gene expression.

### 2.5. RNA Extraction and cDNA Synthesis

The total RNA from each sample was isolated from the IPEC-J2 cells using a Hybrid-R kit (GeneAll^®^ Biotechnology Co., Ltd., Seoul, Korea) according to the manufacturer’s instructions. The purity and quantity of the purified RNA, free of DNA and proteins, was determined at 260/280 nm, using NanoDrop 8000 (Thermo Scientific, Waltham, MA, USA). RNA was reverse-transcribed to cDNA, using the QuantiTect Reverse Transcription Kit (Qiagen, Germany) according to the manufacturer’s instructions, and the resultant cDNA was stored (at −20 °C) until used.

### 2.6. Gene Expression Analysis (qPCR)

Real-time PCR was performed using CFX Manager Software (CFX Manager version 2.0, BioRad, Hercules, CA, USA) in a 10 μL reaction volume containing 1 × iQ™ SYBR^®^ Green Supermix (BioRad), 0.5 μM forward and reverse primers and 40 ng/μL of cDNA. GAPDH was used as a reference gene for internal control. Each assay included a negative control without a cDNA template. All reactions were performed in triplicate. The experimental protocol consisted of the initial denaturation at 95 °C for 5 min, followed by amplification including 40 cycles of 3 steps (denaturation at 94 °C for 30 s, annealing at 60 °C for 30 s, extension at 72 °C for 30 s), and final extension at 72 °C for 15 min, followed by melting curve analysis to confirm the amplification of a specific product. Relative normalized expression was calculated using the 2^−ΔΔCT^ method. Results of the gene expression experiment conducted in triplicate were expressed as mean ± standard deviation (SD). Primers for the mRNA expression analysis are listed in Table 2. For gene expression, a data-visualization heat map was constructed using GraphPad Prism 9.0.0 software.

### 2.7. Dot Blot Analysis

Culture supernatants were collected at indicated times for IL-8 analysis, using the dot blot method. Briefly, 400 μL of each supernatant was lyophilized, and the obtained pellet was then resuspended in 40 μL of sterile distilled water. A total of 2 μL of IL-8 protein (1 ng/uL) (Antibodies-online GmbH, Aachen, Germany) and 2 μL of concentrated sample supernatants were applied to dry nitrocellulose membranes. Once the membranes had dried completely, they were subsequently blocked with 5% BSA in Tris-buffered saline containing Tween-20 (0.05%) for 1 h, at room temperature. The membranes were then incubated in 1% BSA in Tris-buffered saline containing Tween-20 (0.05%) containing rabbit anti-IL-8 polyclonal antibody (Antibodies-online GmbH, Germany) at a 1:300 dilution, for 1 h at room temperature. The membranes were then washed three times with Tris-buffered saline containing Tween-20 (0.05%) (washing buffer) and then incubated at room temperature with CF^®^ 770 goat anti-rabbit IgG conjugated antibody (Bio-Connect, Huissen, The Netherland) at a 1:2000 dilution for 1 h, in 1% BSA in Tris-buffered saline containing Tween-20 (0.05%). The membranes were washed three times with washing buffer and scanned using an Odyssey Scanner (LI-COR Biosciences, Lincoln, NE, USA). 

### 2.8. Statistical Analysis

The results of the gene expression are expressed as the mean and standard deviation (SD) of two independent experiments. Significant differences were determined using GraphPad Prism 9.0.0 software by one-way analysis of variance (ANOVA) followed by Dunnett’s multiple comparison. The level of significance was set at *p*-value ≤ 0.05 considered significant (*), *p*-value ≤ 0.01 considered very significant (**), *p*-value ≤ 0.001 (***)–0.0001 (****) considered extremely significant, and ns considered not significant.

## 3. Results

### 3.1. The Effect of EPS Isolated from L. reuteri L26 (EPS) and L. reuteri L26 (L26) on the Expression of Genes Encoding Pro-Inflammatory Cytokines in IPEC-J2 Cells Challenged with S. Typhimurium (ST)

The aim of this study was to compare the pathogenic and probiotic microbe-mediated responses at cellular level by individually exposing the porcine jejunal-cell-line to live pathogenic bacteria, *S.* Typhimurium (ST), as well as to the live probiotic bacterial-strain *L. reuteri* L26 (L26) and its purified exopolysaccharide. More precisely, we focused on the study of the immune response of IPEC-J2 cells treated with EPS isolated from *L. reuteri* L26 (EPS), or live L26, or challenged with live ST, by performing quantitative real-time PCR (qPCR) to determine the mRNA expression of some pro-inflammatory cytokines. We also monitored the immune response of IPEC-J2 pretreated with EPS or L26 before ST infection. Using relative quantification, we evaluated the gene expression of some important pro-inflammatory cytokines (TNF-α, IL-6, IL-8) and the regulatory cytokine TGF-β. In addition to mRNA expression, the dot blot method was performed to determine the protein secretion of IL-8 in cell supernatants. Incubation of IPEC-J2 cells with ST led to increased gene expression in all studied pro-inflammatory cytokines. The relative level of IL-8 was most affected by the L26 treatment, with an approximately fourfold increase in mRNA expression in the IPEC-J2 cells; however, the pretreatment of cells with L26 resulted in the suppression of IL-8 (*p* < 0.001) compared with cells exposed to ST alone (Figure 1). 

A similar trend was observed in the case of TNF-α expression, when the relative level of this cytokine was significantly increased in the IPEC-J2 cells treated with L26 (*p* < 0.0001) compared with ST infection; meanwhile, treatment with *S.* Typhimurium alone, increased mRNA expression of TNF-α approximately twofold (Figure 1). Although this up-regulation of TNF-α induced by probiotics in the case of pretreatment was lower, it was without statistical significance. The administration of EPS alone to the IPEC-J2 cells significantly suppressed IL-8 expression (*p* < 0.0001) even further; as with the L26 pretreatment, the EPS was able to suppress the up-regulation induced by ST (*p* < 0.05; Figure 1). A more pronounced anti-inflammatory response was observed for IL-6, with an extremely significant down-regulation (*p* < 0.0001) in all experimental groups, compared with the ST challenge. The anti-inflammatory effect was manifested by a significantly increased TGF-β mRNA expression level in the L26 experimental group as compared to the ST-challenged group (*p* < 0.0001) (Figure 1). In addition to gene expression, we collected cell supernatants and detected the presence of IL-8 protein by the dot blot method. We found high concentrations in cell supernatants obtained from all experimental groups. As a positive result, the presence of IL-8 protein in the cell supernatants was evaluated by comparison with the positive control (blotted IL-8 protein) and with the negative control (cell supernatants achieved from untreated IPEC-J2) (Figure 2. The dot blot method was evaluated subjectively and semi-quantitatively, according to the intensity of staining on the nitrocellulose membrane. The intensity of detected chemiluminescence of each sample correlated with the results of the gene expression analysis in each experimental group. Spots on the nitrocellulose membrane showed positive results with different color tints, rather than a negative result (without reaction).

### 3.2. The Effect of EPS Isolated from L. reuteri L26 (EPS) and L. reuteri L26 (L26) on Expression of Genes Related to TLR Cascade in IPEC-J2 Cells Challenged with S. Typhimurium (ST)

We aimed to evaluate the expression of TLR4 and TLR5 genes in IPEC-J2 cells, as epithelial TLR expression appears to be key in the host defense against bacterial infection [17]. The relative levels of TLR4 and TLR5 mRNA were most increased by the Salmonella challenge, approximately fourfold in the case of TLR4 and twofold in the case of TLR5 (Figure 3). TLR5 mRNA expression was significantly reduced (*p* < 0.001) in cells treated with EPS, as well as in cells pretreated with EPS (*p* < 0.001) or L26 (*p* < 0.01), compared with cells challenged with ST (Figure 3). Similar results were observed for the mRNA transcriptional level of TLR4 in cells treated with EPS (*p* < 0.05), or pretreated with EPS (*p* < 0.0001), compared with cells challenged with ST (Figure 3). Although the relative level of TLR4 mRNA was significantly down-regulated (*p* < 0.0001) in L26-treated cells compared with the ST infection, the TLR5 mRNA expression in L26-treated cells was without significance (Figure 3).

## 4. Discussion

As enterocytes in the presence of probiotic bacteria are able to secrete various cytokines in a strain-dependent manner, studies suggest that the interaction of probiotic bacteria with the intestinal epithelium is a key determinant for cytokine production, and is likely to be an initiating phenomenon in the probiotic immunomodulatory-activity that occurs before encountering immune-system cells [18]. Signaling pathways ensured by PRRs of enterocytes or immune cells in the *lamina propria* of the gut, support the intestine’s protective function by inducing inflammation, which eliminates microbial invaders [8,19]. The binding of a specific ligand to the appropriate TLR receptor initiates dimerization, leading to the recruitment of adapter molecules such as MyD88, TIRAP (Mal), or TRIF, which leads to the activation of the transcription factors NF-κB, AP1, or IRF and to the subsequent up-regulation of inflammatory cytokines, chemokines, various antimicrobial molecules, co-stimulatory molecules, and so on [20]. We demonstrated that the relative level of TLR4 mRNA was most increased by *Salmonella* induction. However, EPS treatment and the pretreatment of IPEC-J2 cells up-regulated the mRNA transcriptional level of TLR4, although compared with the *Salmonella* infection it was still significantly lower (Figure 3 and Figure 4). This TLR4 up-regulation may indicate that probiotic EPS keeps the host in a state of vigilance for pathogens. In the study of Laiño et al. (2016) it was proposed that the molecular diversity of lactic-acid-bacteria (LAB) EPS affects their immunogenicity. They pointed out how different EPSs induce the expression of different negative regulators of the TLR4 signaling pathway, thereby modulating the NF-κB transcriptional pathways and suppressing the production of pro-inflammatory cytokines and chemokines [21]. On the other hand, there are available studies indicating that other types of EPS may contain structural features which TLR can directly recognize, thereby eliciting a stimulatory immune-response [4]. The immune system is able to differentiate between EPS produced by pathogens and by commensal bacteria. While some commensal bacteria exploit the TLR pathway to actively suppress immune reactions, ligands of pathogens trigger inflammation via host TLR [22]. These findings point to strategies by which beneficial bacteria exert their immunomodulatory effect. As we observed, the highest expression of the inflammatory cytokines IL-8 and TNF-α was in the ST and L26 experimental groups (Figure 1), and it is not surprising that TLR5 transcription levels measured by qPCR were up-regulated just in these wells. Several in vivo studies suggest that PRR signaling is required to maintain intestinal homeostasis; nevertheless, various mechanisms are used to suppress this signaling in the healthy gut [23]. TLR signaling in the gut ensures the regulation of immune tolerance to commensals, as well as an immune response to the intestinal pathogens [20]. Innate receptors play a vital role in the balance between the induction and reduction of inflammation in the host, and therefore constant TLR stimulation may be necessary for maintaining intestinal health [24]. TNF-α represents a crucial pro-inflammatory cytokine with pleiotropic functions in gut inflammation, where it is produced by invading immune cells and by the IECs themselves. On the other hand, it is proven that the induction or the inhibition of TNF-α expression by beneficial probiotic LAB might provide either immune-stimulating or immune-suppressant effects for the host [25]. It is known from several studies that LAB are able to increase the production of a range of pro-inflammatory cytokines in addition to TNF-α, such as IFN-γ, IL-8, IL-1β, IL-12 and IL-6, thereby triggering physiological inflammation. Different strains of probiotics can induce TNF-α secretion, by peripheral blood mononuclear cells and dendritic cells increasing the number of TNF-α-producing cells in the *lamina propria* of the gut [26,27,28,29]. We observed significant up-regulation of TNF-α expression in L26-treated IPEC-J2 cells, compared with cells challenged with ST; however, in IPEC-J2 cells pretreated with either EPS or L26, we did not observe overexpression. A similar ability of lactobacilli to induce increased TNF-α expression has been observed in other studies. A study by Lee et al. (2016), showed that *Lactobacillus plantarum* K55-5 and *Lactobacillus sakei* K101 were able to significantly increase the production of cytokines such as IL-10, IL-12, IFN-γ, and TNF-α in immunosuppressed-mice models [25]. In another study of the immunobiotic potential of LABs, it was proven that the probiotic lactobacilli strains were able to increase TNF-α and IL-8 levels, which were maintained for 24 h [30]. These findings suggest that, although treatment of cells with live LAB had an immune-stimulatory effect on IPEC-J2 cells, in the case of pretreatment they were able to suppress the *Salmonella* which caused up-regulation. In this study, we evaluated the expression level of IL-8, also known as CXCL8, a potent neutrophil chemoattractant in IPEC-J2 cells. We observed in group of cells challenged with *S.* Typhimurium, that the relative expression level of IL-8 was increased, but the increase in expression was also induced by treatment of the cells with the probiotic strain L26 alone. Although L26 treatment elicited the highest level of relative IL-8 mRNA expression, we observed a significant down-regulation of IL-8 in cells pretreated with L26 prior to the *S.* Typhimurium challenge (Figure 1). Likewise, Skjolaas et al. (2007) also demonstrated that pretreatment of IPEC-J2 cells with the probiotic strain *Bacillus licheniformis* led to a significant decrease in the secretion of IL-8 after subsequent exposure to *S.* Typhimurium; however, no difference was observed with previous exposure to *L. reuteri* [31]. This demonstrates that different probiotic bacteria can have different effects on the host response to *S.* Typhimurium. Similar results were obtained in the study of Kanmani and Kim (2020), where they found that immunobiotic strains attenuate the *Salmonella*-induced pro-inflammatory response in human intestinal-epithelial-cells. Their study showed that HT-29 cells treated with *Salmonella* up-regulated the expression of some relevant pro-inflammatory cytokines such as IL-8, IL-6, IL-1β, and MCP-1, but that these increases were differentially altered in the case of lactobacilli-pretreated cells [32]. These results indicate that the pretreatment of cells with probiotics has a significant protective effect on IECs, and that he infection does not progressively develop at the level of gene expression of inflammatory cytokines. We demonstrated that pretreatment of IPEC-J2 cells with either the probiotic strain L26 or EPS reduced IL-6 expression induced by the *Salmonella* challenge, proposing their potential anti-inflammatory effect on IECs in vitro. We observed a similar down-regulating effect in a group of cells treated with L26 or EPS, alone. However, despite the recognized importance of IL-6, its mechanisms of action on the intestinal barrier are not well known and, in particular, its effect on the mucosal barrier is still unclear [33]. It is generally known that the transforming growth factor (TGF)-β is abundant in the intestine, and it is also produced by epithelial cells. TGF-β was identified as a negative regulator of NF-κB activation in gut inflammation [34]. In a study by Huang et al. (2015), it was revealed that the *S.* Typhimurium-induced up-regulation of NF-κB in the human intestinal Caco-2 cell line was altered by *L. acidophilus*-induced TGF-β/microRNA-21 (MIR21) expression. They propose that the TGF-β/MIR21 expression could serve as an anti-inflammatory evaluation tool for another strains of LAB, but it requires further study in this field [35]. Our current study demonstrates that TGF-β was significantly up-regulated in response to L26 treatment in IPEC-J2 cells. These results are in line with a previous in vivo study, where the orally administered probiotic *Enterococcus faecium* EF1 was able to increase the concentrations and mRNA expression levels of TGF-β1 and TNF-α in the jejunal mucosa of newborn piglets. Their findings revealed that this LAB strain effectively modulates the expression of TLRs and activates innate immunity in piglet jejunal mucosa [24]. 

## 5. Conclusions

In summary, the results from the present study show immunoregulatory functions by the pretreatment of porcine intestinal-epithelial IPEC-J2 cells with the probiotic strain *L. reuteri* and its component exopolysaccharides, compared with *Salmonella* infection. The infection of IPEC-J2 by *S.* Typhimurium was confirmed by a pro-inflammatory transcriptional answer in these cells. We found up-regulation of mRNA expression of the pro-inflammatory cytokines TNF-α, IL-8, and IL-6, and the TLR4, TLR5 signaling pathways in infected cells. Although *L. reuteri* was able to significantly up-regulate the expression levels of TNF-α and IL-8, in the case of pretreatment this up-regulation was reduced. We therefore assume that *L. reuteri* L26 has both pro- and anti-inflammatory properties. An interesting finding was that EPS treatment of cells alone had immunosuppressive potential, since the mRNA levels of observed pro-inflammatory cytokines were not increased. This immunosuppressive property was also manifested in pretreated cells with EPS. Pretreatment of cells with either live lactobacilli or their purified EPS was able to suppress the infection-induced up-regulation of the noted pro-inflammatory cytokines. Based on these observations, the present study suggests potential immunoregulatory functions for the studied lactobacilli and its exopolysaccharides. However, these observations were based on the use of cell lines as experimental models, and do not necessarily represent the true situation in vivo. As the health effect of lactobacilli is strain-dependent either in the direction of immune-stimulation or immune-suppression, additional ex vivo co-cultivation studies and in vivo studies are required to confirm our hypothesis.

## Figures and Tables

**Figure 1 life-12-01955-f001:**
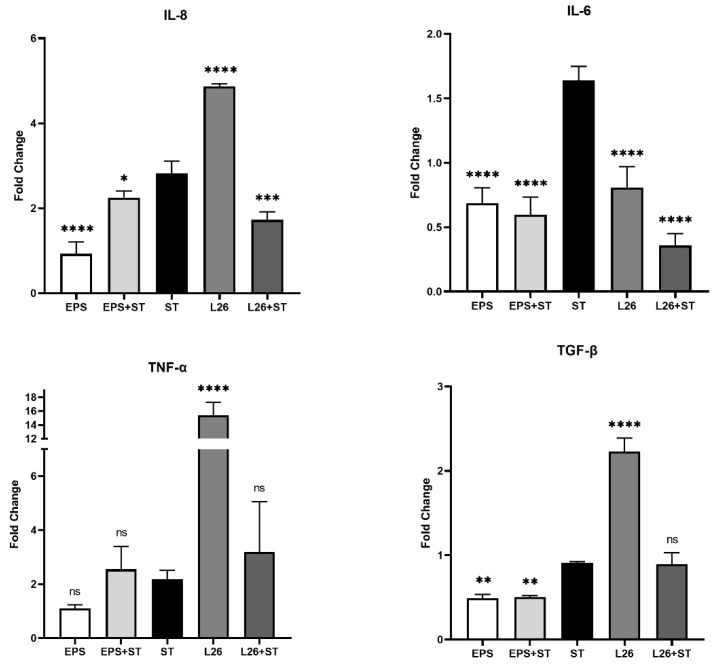
The effect of EPS produced by *L. reuteri* strain L26 Biocenol^TM^ (EPS) and *L. reuteri* strain L26 Biocenol^TM^ (L26) on expression of pro-inflammatory genes in IPEC-J2 cells and on expression of anti-inflammatory gene TGF-β in IPEC-J2 cells. Results are expressed as mean ± SD. The statistical significance of the differences was evaluated using Dunnett’s multiple comparison test. Results were significantly different compared with *Salmonella* Typhimurium CCM 7205 (ST) (* *p* < 0.05; ** *p* < 0.01; *** *p* < 0.001; **** *p* < 0.0001; ns: not significant). The error bars indicate standard deviations.

**Figure 2 life-12-01955-f002:**
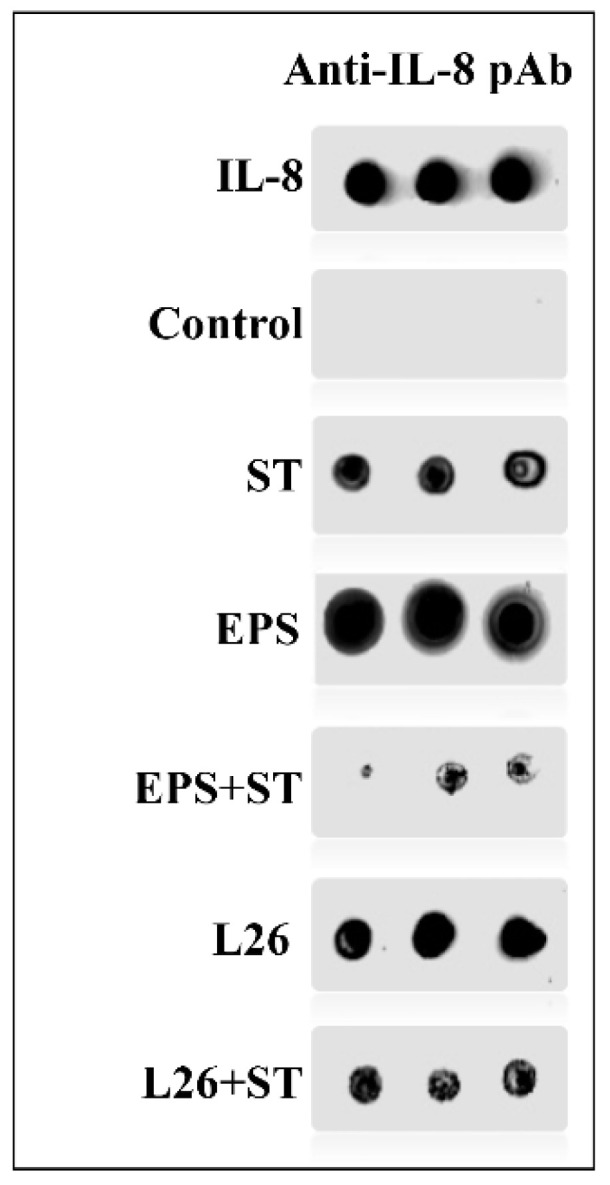
Dot blot analysis of culture supernatants of each experimental group performed in triplicate, spotted onto nitrocellulose membrane for IL-8 detection. Supernatants from individual experimental groups expressing IL-8 on nitrocellulose membrane. Detection was performed with rabbit anti-IL-8 polyclonal antibody obtained from Antibodies-online GmbH, and with CF^®^ 770 goat anti-rabbit IgG conjugated antibody, obtained from Bio-Connect. The membranes were scanned in an Odyssey Scanner.

**Figure 3 life-12-01955-f003:**
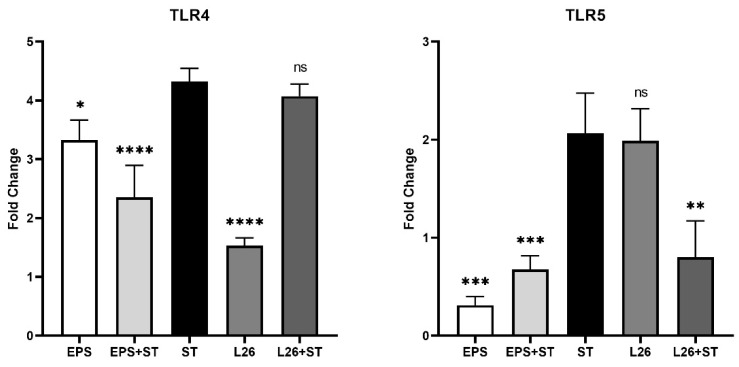
The effect of EPS produced by *L. reuteri* strain L26 Biocenol^TM^ (EPS) and *L. reuteri* strain L26 Biocenol^TM^ (L26) on expression of genes related to TLR cascade in IPEC-J2 cells. Results are expressed as mean ± SD. The statistical significance of the differences was evaluated using Dunnett’s multiple comparison test. Results were significantly different compared with *Salmonella* Typhimurium CCM 7205 (ST) (* *p* < 0.05; ** *p* < 0.01; *** *p* < 0.001; **** *p* < 0.0001; ns: not significant). The error bars indicate standard deviations.

**Figure 4 life-12-01955-f004:**
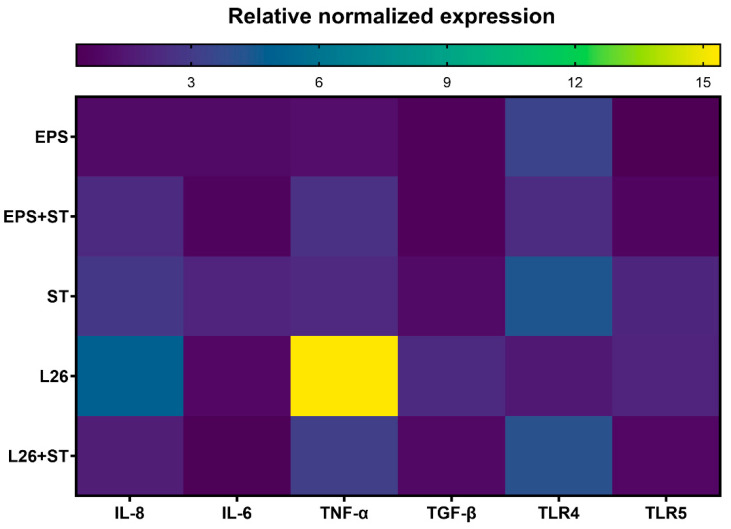
Illustration of gene-expression data by heat map. Heat map showing gene expression in the IPEC-J2 cells of each experimental group. The bar shows relative normalized expression values, with the corresponding color code.

**Table 1 life-12-01955-t001:** Table of cell treatments.

Type of Cell Treatment	Experimental Design	Abbreviation
Control	IPEC-J2 cells were without any treatment.	_
ST infection	IPEC-J2 cells were challenged with *Salmonella* infection for 1 h. After this incubation period, for the purpose of killing the remaining extracellular bacteria gentamicin was added, and the cells were further incubated for the next 3 h in the presence of 100 ug/mL of gentamicin.	ST
EPS/L26 treatment	IPEC-J2 cells were treated with EPS 100 ug/mL for 4 h.	EPS
	IPEC-J2 cells were treated with *L. reuteri* L26 for 5 h.	L26
EPS/L26 pretreatment	IPEC-J2 cells were pretreated with EPS 100 ug/mL for 4 h, and subsequently challenged with *Salmonella* infection for 1 h. After this incubation period, for the purpose of killing the remaining extracellular bacteria gentamicin was added, and the cells were further incubated for the next 3 h in the presence of 100 ug/mL of gentamicin.	EPS + ST
	IPEC-J2 cells were pretreated with *L.reuteri* L26 for 5 h, and subsequently challenged with *Salmonella* infection for 1 h. After this incubation period, for the purpose of killing the remaining extracellular bacteria gentamicin was added, and the cells were further incubated for the next 3 h in the presence of 100 ug/mL of gentamicin.	L26 + ST

**Table 2 life-12-01955-t002:** PCR primer used in this study.

Genes	Forward Primer (5′ → 3′)	Reverse Primer (5′ → 3′)	T_m_ (°C)	Reference
GAPDH	ACT CAC TCT TCT ACC TTT GAT GCT	TGT TGC TGT AGC CAA ATT CA	60	[14]
TLR4	CTC TGC CTT CAC TAC AGA GA	CTG AGT CGT CTC CAG AAG AT	60	[15]
TLR5	TTT CTG GCA ATG GCT GGA CA	TGG AGG TTG TCA AGT CCA TG	60	[15]
IL-6	TGG ATA AGC TGC AGT CAC AG	ATT ATC CGA ATG GCC CTC AG	60	[15]
IL-8	CGC ATT CCA CAC CTT TCC ACC CC	TCC TTG GGG TCC AGG CAG ACC	60	[16]
TNF-α	CGA CTC AGT GCC GAG ATC AA	CCT GCC CAG ATT CAG CAA AG	60	[12]
TGF-β	CAC GTG GAG CTA TAC CAG AA	TCC GGT GAC ATC AAA GGA CA	60	[15]

## Data Availability

All the data associated with this research is included in this article. Any further information is available upon reasonable request.

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
