# Peer review of "Immunomodulatory Effect of Lactobacillus reuteri (Limosilactobacillus reuteri) and Its Exopolysaccharides Investigated on Epithelial Cell Line IPEC-J2 Challenged with Salmonella Typhimurium"

_life, 2022, doi:10.3390/life12121955_

Round 1

Reviewer 1 Report

This manuscript introduced that Lactobacillus reuteri (Limosilactobacillus reuteri) and its exopolysaccharides can immunomodulate the inflammation of IPEC-J2 cell line caused by Salmonella Typhimurium. The following are the questions in this manuscript:

1.      Part Abstract, the research background is too many and the demonstrated study results are not enough.

2.      In the text, the strain name, like Lactobacilli, should use in Latin.

3.      Lines 52-53, supplementing references.

4.      Line 84, delete the second “was”.

5.      Line 118, Table 2 appears before Table 1.

6.      Table 2, experimental design should be revised, there are many grammar questions. Additionally, for EPS/L26 pretreatment, what is the purpose of adding gentamicin?

7.      Figure 1, what is the unit in Y axis? And how to calculate the relative normalized expression? Same to Figure 2.

8.      Part 3.1, why the probiotic stimulated the secretion of proinflammatory cytokine, supplementing discussion.

9.      Figure 2, for TLR4, the expression level in the EPS treatment group is higher than that in EPS+ST, what is the reason?

10.   There are also a few formatting problems, such as line 47 “S. enterica” should be “S. enterica”; line 151 “1 ng/ul should be “1 ng/uL”; line 170, etc.

11.   There is problem with the chart format and need to be corrected, such as line 147-148, etc.

12.   In the result part, the language description is too single, and further concise summary is needed.

13.   In the discussion part, I think too much introduction to the functions of various inflammatory factors should be deleted.

14.   The conclusion part needs to be supplemented with some experimental data to increase its persuasiveness, such as the change of gene expression level of inflammatory factors.

15.   There was a problem with the reference format, such as line 403, etc.

Reviewer 2 Report

1.  Line 190-191: A more pronounced anti-inflammatory response was observed for IL-6, why did IL-6 not perform protein level verification.

2.  The control group selection problem? Other probiotics that are already known should be added as a positive control and then compared together.

3.  The references format needs to be unified, and some references lack page numbers. Missing page numbers at lines 411,418,421.

4.  In addition to the above results, whether this study is verified in animals.

Reviewer 3 Report

Comments to the Author (Life - Special Issue “Natural Substances in Nutrition and Health of Animals”): 

The manuscri entitled "Immunomodulatory effect of Lactobacillus reuteri (Limosilactobacillus reuteri) and its exopolysaccharides investigated on epithelial cell line IPEC-J2 challenged with Salmonella Typhimurium" by Kiššová, Z. et al., is focused on effect of application of probiotic strain Limosilactobacillus reuteri L26 BiocenolTM and its components – exopolysaccharides, compared to Salmonella infection in in vitro conditions carried out on IPEC-J2 cell line. Pretreatment of IPEC-J2 cells with either live lactobacilli or its EPS down-regulated the Salmonella caused up-regulation of expression of the studies pro-inflammatory cytokines. The manuscript is scientifically interesting, the work is well and clearly written, but there were a few mistakes (described below). I recommend the manuscript for publishing after making the following concerns.

Comments and Suggestions:

Throughout the text – please and add a space after a sign „<“ (e.g. P < 0.05)

Throughout the text please correct the registered trademark symbol ® - use as a superscript

You used the explanation for PBS in P3/L103, but you already used the abbreviation in P2/L78. Please fix it

P1/L32 - The abbreviation EPS has already been used in the abstract, there is no need to write in brackets again

In the Material and Methods the abbreviation "hours" is misspelled several times, there is no space between the numbers and the abbreviate hour symbol "h". The same mistake happened with abbreviation "degree Celsius". Please correct the mistakes, fill the gaps.

P2/L67 - 24h - fill the gap
P2/L77 - 37°C - fill the gap
P3/Section 2.4.1 - Please clarify why did you use different incubation times for EPS (4 h) and for live lactobacilli (5 h)?

P3/L92; L102; L105; L108; L118; L121; L122 - 37°C - fill the gap
P3/L93; L96 – please, use „h“ instead of „hours“

P7/Table2 – please fill the gaps between the numbers and the abbreviate hour symbol "h"

P3/L97 - Please clarify what the used cell medium contained, what do you mean by the term "without supplementation" in the following sentence - "24 hours prior to experiment the cell medium was changed to DMEM/F-12 without supplementation"

P3/L106 - Please add information which bacterial cultures were diluted at „4 h cultures or overnight cultures of bacteria“

P3/L107 – Please correct „... multiplicity of bacteria (MOI)...“ to „.... multiplicity of infection (MOI)...“

P4/L134 - Please correct „CFX96Manager Software“ to „ CFX Manager Software“

P5/L197 - Please change the sentence: „The anti-inflammatory effect was manifested in... to „ The anti-inflammatory effect was manifested by significantly increased TGF-β mRNA expression level in L26 experimental group as compared to ST challenged group (p ˂ 0.0001).“

P5/L204 - Please clarify how you evaluated the dot blot method. If no quantitative analysis was performed, add information about evaluation (Was there used subjective evaluation according to the intensity of staining on NC membrane?) – Please clarify it in the next sentence "These results obtained from cell supernatants evaluated with dot blot method were in agreement with the results obtained in mRNA expression."

In the Discussion section, please try to reformulate the sentences that begin with the wording "In this study.....". Use a different wording to avoid repeating yourself.

P9/L263; L288     - please rewrite „In this study...“
P10/L323 - please rewrite „In this study...“
